# EVOO Polyphenols Relieve Synergistically Autophagy Dysregulation in a Cellular Model of Alzheimer’s Disease

**DOI:** 10.3390/ijms22137225

**Published:** 2021-07-05

**Authors:** Manuela Leri, Andrea Bertolini, Massimo Stefani, Monica Bucciantini

**Affiliations:** 1Department of Biomedical, Experimental and Clinical Sciences, University of Firenze, 50134 Firenze, Italy; manuela.leri@unifi.it (M.L.); monica.bucciantini@unifi.it (M.B.); 2Department of Surgical, Medical, Molecular Pathology and Critical Care Medicine, University of Pisa, 56126 Pisa, Italy; a.bertolini2@student.unisi.it

**Keywords:** autophagy, polyphenols, Alzheimer’s disease, oligomers

## Abstract

(1) Background: Autophagy, the major cytoplasmic process of substrate turnover, declines with age, contributing to proteostasis decline, accumulation of harmful protein aggregates, damaged mitochondria and to ROS production. Accordingly, abnormalities in the autophagic flux may contribute to many different pathophysiological conditions associated with ageing, including neurodegeneration. Recent data have shown that extra-virgin olive oil (EVOO) polyphenols stimulate cell defenses against plaque-induced neurodegeneration, mainly, through autophagy induction. (2) Methods: We carried out a set of in vitro experiments on SH-SY5Y human neuroblastoma cells exposed to toxic Aβ_1–42_ oligomers to investigate the molecular mechanisms involved in autophagy activation by two olive oil polyphenols, oleuropein aglycone (OleA), arising from the hydrolysis of oleuropein (Ole), the main polyphenol found in olive leaves and drupes and its main metabolite, hydroxytyrosol (HT). (3) Results: Our data show that the mixture of the two polyphenols activates synergistically the autophagic flux preventing cell damage by Aβ_1–42_ oligomers., in terms of ROS production, and impairment of mitochondria. (4) Conclusion: Our results support the idea that EVOO polyphenols act synergistically in autophagy modulation against neurodegeneration. These data confirm and provide the rationale to consider these molecules, alone or in combination, as promising candidates to contrast ageing-associated neurodegeneration.

## 1. Introduction

Alzheimer’s disease (AD), characterized by the development of neuronal amyloid-β (Aβ_1–42_) plaques and tau neurofibrillary tangles [1], accounted for 60 to 80% of dementia cases in 2015, affecting 46.8 million people worldwide [2], a number expected to double every 20 years, reaching 74.7 million in 2030 and 131.5 million in 2050 [3]. Accordingly, dementia pandemic urgently requires effective treatments. However, the great majority of potential disease-modifying therapies clinical trials for AD yielded substantially negative results over the past 20 years. These trials tested a variety of treatments, including antioxidants, statins, non-steroidal anti-inflammatory drugs, estrogens and nerve growth factor gene therapy. Furthermore, despite genetic and molecular evidence pointing to Aβ_1–42_ as a key player in AD pathogenesis [4], most trials with anti-amyloid therapies failed to provide evidence of treatment efficacy, suggesting that the anti-amyloid strategy should be abandoned [5] and new therapeutic avenues searched. Numerous longitudinal studies using several AD biomarkers indicated that AD develops decades before symptoms appearance [6,7], recommending the usefulness of preventive multi-target treatments aiming at hindering or delaying disease onset.

Recent epidemiological studies support the efficacy of the Mediterranean diet not only against cardiovascular pathologies, cancer, type 2 diabetes and the metabolic syndrome but also against the cognitive decline associated with ageing [8]. Past research has dedicated special emphasis to the pathways involved in AD onset and progression, including amyloid precursor protein processing, Aβ_1–42_ peptide and tau protein aggregation, autophagy impairment, synaptic derangement, oxidative stress and neuroinflammation [9]. Several data highlight the ability of various polyphenols, including phenolic components of red wine and olive oil, to counteract amyloid aggregation and aggregates toxicity in different aging-associated diseases, including neurodegeneration [10,11]. Yet, the benefits of plant polyphenols seem to go well beyond their anti-amyloid and antioxidant power. However, although the claimed benefits of plant polyphenols against neurodegeneration have been supported by many preclinical studies, both in vitro and with animal models, so far, those benefits have not been convincingly confirmed in humans by well-designed clinical trials Indeed, the low bioavailability of orally ingested polyphenols greatly limits their therapeutic use, particularly in those organs distant from the gastrointestinal tract. Nevertheless, there is evidence from animal studies and clinical trials that several polyphenols can cross the BBB and reach brain parenchyma [12,13].

Extra Virgin Olive Oil (EVOO) is rich in polyphenols, whose concentration can be as high as 600–800 mg/kg. Oleuropein aglycone (OleA) and hydroxytyrosol (HT), its main metabolite, are the most abundant polyphenols in olive oil, where they account for over 80% of the total phenolic fraction. Recent data indicate that OleA interferes with APP processing [14] and amyloid aggregation of amylin, Aβ_1–42_ peptides, α-synuclein, transthyretin and tau protein, avoiding the growth of toxic oligomers both *in vitro*, in a Tg strain of Aβ-expressing C. elegans and in a TgCRND8 mice, a model of Aβ_1–42_ plaque deposition. In the latter, a strong protection by OleA was reported: Tg mice fed with a normal diet supplemented with OleA showed a dose-dependent protection against cognitive deterioration, as compared with normally fed littermates. At a tissue level, these mice displayed a significant improvement of synaptic functions, plaque load, neurogenesis and neuroinflammation, but not a significant reduction of the oxidative stress. Stimulations of cell defenses against plaque-induced neurodegeneration and a remarkable activation of the autophagic flux, normally impaired in neurodegeneration [13,15], were also reported. Taken together, these data confirmed the multi-task activity of this molecule. Similar results were reported in the same mice fed with HT, indicating the latter as the active component of the OleA molecule [16].

The individual OleA and HT beneficial effects against Aβ_1–42_ deposition and cytotoxicity in the same animal model does not exclude the possibility that these specific effects could be produced, with different efficacy, by both molecules together. In this case, we could expect a protection potentiation by a mixture of both molecules compared to that resulting from each single component.

To address this possibility, we treated human neuroblastoma SH-SY5Y cells with a mixture of both OleA and HT, before exposing them to toxic Aβ oligomers. We evaluated the ability of the aggregates to bind the cell membrane and measured various biochemical/cellular parameters associated with cell sufferance, including, oxidative stress, mitochondrial function and autophagy.

## 2. Results

### 2.1. OleA/HT Mixtures Trigger Autophagy

Many polyphenols, including OleA, display neuroprotection by triggering an autophagic response both in cultured cells and in animal models. As previously reported, a 50 μM OleA concentration potently activates autophagy after 4 h of cell treatment [17,18]. To investigate the pro-autophagic protective effect of OleA against aggregates insults, we exposed our cells for 4 h and 24 h to a 75 μM OleA or HT concentration, by comparison, before the addition of Aβ_1–42_ aggregates (Appendix A). We observed a significantly reduction of ROS production in both treatments, probably due to the antioxidant activity of both molecules (Appendix A). Nevertheless, confocal microscopy analysis showed the presence of amyloid aggregates on neuroblastoma cell surface also after the pre-treatment with the single polyphenols (Appendix A). Then, we assessed the impact of the two polyphenols on autophagic activity and, considering that, in the intestine, the microflora decomposes a large fraction of OleA to HT, we sought to assess, in parallel, whether OleA/HT mixtures displayed synergistic activity on the autophagic path respect to the same amounts of each single component. Initially, we treated SH-SY5Y cells with OleA or HT, at a 75 μM concentration either separately (Appendix A) or in combination at different (2/1; 1/1; 1/2) OleA/HT molar ratios (Appendix A). We used the Cyto-ID^®^ staining probe, which selectively recognizes the autophagic vacuoles, to determine the most effective condition to activate autophagy. OleA and HT, alone or in combination, were previously assessed for safety to cells (Appendix A). The confocal microscopy images showed an overtime protracted increase of autophagosome formation in cells subjected for 4 h and 24 h to both polyphenols in 1:1 (37.5 μM:37.5 μM) molar ratio, compared to autophagosome levels found in cells treated with each polyphenol administered individually or in combination in a 1:2 or 2:1 molar ratio (Appendix A). We therefore used the 1:1 molar ratio (MIX) in all subsequent experiments.

Once assessed the optimal OleA:HT ratio in terms of autophagy activation, we analyzed specifically the time-course of this process. The Cyto-ID^®^ fluorescence was increased in cells treated with the MIX for 0.5 h up to 24 h (Figure 1A). The expression levels of autophagy main markers were also analyzed (Figure 1B–E). The first investigated marker was the phosphorylation level of the ribosomal protein S6, a key downstream substrate of the main suppressive regulator of autophagy, mTOR. We observed a reduction of the phosphorylation level of S6 (pS6) in cells treated with the MIX for up to 24 h, as compared to untreated control cells (Figure 1B). Under our experimental conditions we found that S555 phosphorylation in ULK1 was increased, particularly after 4 h, in cells treated with the MIX (Figure 1C), indicating the involvement of the AMPK pathway (in agreement with previous results concerning autophagy activation by the sole OleA [18]).

Beclin-1, a key regulator of autophagosome formation, was also increased in cells treated for 1 h to 6 h with the MIX (Figure 1D). We also investigated the LC3-II/LC3I ratio, a value proportional to the conversion of LC3-I to LC3-II (the lipidated form) and indicative of autophagosome formation. We found that the 1:1 MIX induced a progressive increase of the LC3-II/LC3I ratio, that peaked at 24 h respect to untreated control cells (Figure 1E), in accordance with the Cyto-ID^®^ data. Moreover, we observed a significant reduction of the autophagosome cargo protein p62 expression level, a marker of the autophagic degrading phase, specifically at the earlier times of treatment (0.5 h, 1 h and 6 h) (Figure 1F). Overall, our findings confirm that the MIX triggers the autophagic activation process in SH-SY5Y cells.

### 2.2. Cells Pre-Treatment with the MIX Prevents Aβ Oligomers Cytotoxicity

After, we investigated whether cell treatment with the MIX affected the cytotoxicity, in terms of ROS production, of Aβ_1–42_ pre-fibrillar aggregates. To this purpose, we added the polyphenols mixture to the cell culture medium for 4 h or 24 h and then exposed the same cells for further 24 h to Aβ_1−42_ solutions (2.5 μM, monomer concentration) enriched of oligomers (Ol) or fibrils (Fib) obtained by aging the peptide in batch for 24 or 72 h under aggregation conditions, respectively. 

The redox status of cells treated with the MIX was investigated by using the ROS-sensitive fluorescent probe CH2-DCFDA. We found that oligomeric and fibrillar Aβ_1−42_ aggregates increased ROS production in SH-SY5Y cells by about 120 ± 10% and 168 ± 16%, respectively (Figure 2B). Moreover, cell treatment with the MIX reduced the oxidative stress in cells exposed to the oligomers, with a ROS level comparable to that measured in untreated cells (Figure 2A). These results led us to conclude that the presence of the MIX in the cell medium significantly reduces the adverse effects provided by the oligomeric aggregates, avoids ROS production and ROS-induced damage whereas the toxicity of the fibrillar samples, milder by itself, was not completely abolished. 

We also investigated whether the MIX interfered with the presence of Aβ_1−42_ aggregates on the cell surface. It is widely accepted that amyloid aggregates bind to lipid bilayers and that aggregate interaction with cell membranes is a crucial step of amyloid cytotoxicity [19,20]. Considering the reduced toxic effects of Aβ_1–42_ aggregates in cells pre-treated with the MIX, reported above, we analysed any modification of the presence of Aβ_1–42_ amyloids on the surface of these cells. To this aim, we performed confocal microscopy and a sensitized FRET analysis between GM1 fluorescence (by Alexa-488) and immunofluorescence of Aβ_1–42_ aggregates (by Alexa-568) (Figure 2B, bottom panels). In agreement with our previous results [21], we brought to light that both Aβ_1–42_ oligomers and fibrils interacted with ganglioside enriched plasma membrane regions known as raft domains, as indicated by the aggregate-GM1 co-localization (Figure 2B, upper panels) and by FRET efficiency (Figure 2B, bottom panels), and that such interaction correlated with the highest ROS levels observed (Figure 2A). Therefore, we decided to treat cells with the MIX only for 24 h before exposure to Aβ_1−42_ aggregates. We did not detect any oligomer species on the membrane of cells pre-treated with the MIX (Figure 2B), but we did not obtain the same results with the cells exposed to the fibrils sample, confirming the ROS data indicating a modest recovery of redox homeostasis. These results suggest that the reduced presence of oligomeric species on the cell membrane did not depend on the ability of the MIX to interfere with aggregate/membrane interaction but, more likely, resulted from the activation of some cell protective mechanism, notably autophagy. 

### 2.3. The MIX Favours Aβ_1−42_ Oligomers Digestion by Autophagy

Following, we sought to correlate autophagy activation and the significant reduction of oxidative stress described above (Figure 2). To this motive, we first analysed the variation of autophagy markers in SH-SY5Y cells pre-treated with the MIX for 24 h before exposure to Aβ oligomers for additional 24 h. As widely reported in AD pathogenesis, Aβ_1–42_ aggregates induce a build-up of autophagosomes in exposed cells [22,23]. By Western-blotting and densitometric analysis we observed that cell exposure for 24 h to the oligomers resulted in a LC3-I to LC3-II conversion, a specific indicator of autophagosome formation, indicating that misfolded Aβ_1–42_ induced autophagosome formation (Figure 3B). In addition to LC3, other autophagosome markers such as Beclin-1 were also upregulated after cell treatment with the oligomers (Figure 3A). Autophagy is divided into normal flux and block in flux (with autophagosome accumulation), both monitored by detecting p62 level. The latter is implicated in autophagic cargo recognition, incorporated into complete autophagosomes and degraded by autolysosomes [24]. Our western blot and densitometric analysis indicated that p62 levels increased in oligomers-treated SH-SY5Y cells (Figure 3C), suggesting that Aβ_1–42_ oligomers induce the accumulation of autophagosomes which reflects the inhibition of their degradation in our cell model. 

When we carried out the same experiments in cells pre-treated with the MIX, we found out that autophagy impairment induced by Aβ_1–42_ oligomers was reduced. Interestingly, we also observed a decrease of p62 levels in the same cells, confirming a MIX-induced activation of autolysosomes (Figure 3C). Indeed, p62 is a marker of autophagosome-lysosome fusion and usually its levels are reduced in presence of a normal autophagy flux [24]. To confirm these data, we treated our cells with 10 μM chloroquine (CQ), an inhibitor of autolysosome formation [25], for 16 h before the addition of the MIX and Ol. At these conditions, we found oligomers on the surface of cells treated with the CQ+MIX-Ol and Ol, differently from MIX-Ol treated cells (Figure 3D). Taken together, these results indicate that cell-treatment with Aβ_1–42_ oligomers arrested the autophagic flux and induced autophagosome accumulation, but not autophagosome–lysosome fusion. In addition, our data uncovered a correlation between the activation of polyphenol (MIX)-induced autophagy and the reduction of Aβ_1−42_ oligomers on cells surface. These data so suggest that these aggregates are digested by autophagolysosomes induced by the MIX.

### 2.4. The MIX Preserves the Autophagic Flux 

Once assessed the correlation between autophagy activation and recovery of viability, in terms of oxidative stress, in Ol-exposed cells, we investigated the time-dependence of SH-SY5Y cells protection by the MIX. The cells were treated at different times (0.5 h up to 24 h), with Ol in absence or in presence of a pre-treatment with the MIX for 24 h (MIX-Ol). Under these conditions, confocal microscopy imaging confirmed the time-dependent disappearance of Ol on the cell surface (Figure 4A). These results suggest that the induction of autophagy-degradative activity in cells treated with the MIX is an early event starting as early as after 1 h of cell exposure to Ol and reaching a maximum after 4 h. However, we did not obtain the same results when the MIX was added after cell treatment with Ol sample or together with it. In fact, when we exposed our cells to Ol for 24 h and then treated them with the MIX (Ol-MIX), or when we added Ol and MIX to the culture medium at the same time (Ol+MIX), we did not detect any significant reduction of ROS production (Figure 4B). The presence of Ol on the cell surface confirmed the absence of autolysosome induction by the MIX under these conditions (Figure 4C). These data indicate that the polyphenols mixture (MIX) does not revert the injury to the autophagy path induced by Aβ_1–42_ aggregates; rather, the MIX is only able to prevent it, possibly due to the need of a proper time to activate the biochemical modifications, maybe involving epigenetic modulation of the expression of genes involved in autophagy. 

### 2.5. The MIX Affects Mitochondria Functions

Autophagy in AD is also altered for what concerns mitochondria homeostasis. Physiologically, old/altered mitochondria fuse with lysosomes (mitophagy) but under AD conditions, where lysosome activity is reduced, an accumulation of damaged mitochondria has been reported [26]. Mitochondria have been assigned the status of key players contributing to either normal aging or to the onset of AD. Indeed, compromised mitophagy causes decreased energy production as well as increased oxidative stress and Aβ_1–42_ production [27,28]. Accordingly, considering the importance of mitochondria in AD pathogenesis, we assessed the functionality of the latter by MitoTracker CMXRos, which labels functional organelles in red. In addition, the mitochondrial membrane potential was assessed by JC-1, a cationic dye that accumulates in energized mitochondria with a shift from green to red emission. We found out that cell pre-treatment with the MIX prevented the reduction of functionality of these organelles induced by cell exposure to Aβ_1−42_ aggregates. In fact, cell exposure to oligomers induced a significant organelle damage (reduction of red fluorescence) in both assays (Figure 5A,B), confirming that Aβ_1−42_ aggregates heavily injured these organelles. The latters were rescued by cell pre-treatment with the MIX, as indicated by the red signals, comparable to those obtained with control cells (Figure 5A,B). 

### 2.6. Effects of the MIX on the Cross-Talk between Autophagy and the Ubiquitin Proteasome System

Considering the increasingly reported importance of the cross-talk between autophagy and ubiquitin-proteasome system (UPS) in AD pathogenesis [29], we sought to investigate the effect of the MIX on the UPS system. To do this, we pre-treated SH-SY5Y cells with the proteasome inhibitor MG132 before treating them with the MIX and subsequently exposing them to Aβ_1−42_ oligomers (Ol). In the light of the role played by p62 in this cross-talk [30], we evaluated the expression of this protein by confocal microscopy (Figure 6). We found that cell treatment with Ol and MG132 increased p62 levels, in accordance with previous data [31]. However, pre-treatment of these cells with MIX significantly reduced p62 expression, suggesting a possible involvement of UPS on the protection induced by MIX treatment.

## 3. Discussion

In the ageing and neurodegeneration context, nutrition is attracting increasing interest. Diet is an important factor for human health and can no longer be considered simply nutrition; rather, in the light of recent advances in research, especially nutrigenomics, it has been shown to be intimately linked, via evolution and genetics, to cell health. In fact, it improves the efficiency of homeostatic systems in cells/tissues including proteostasis and redox equilibrium through modulation of autophagy/apoptosis, detoxification processes, and appropriate gene responses [9]. In particular, the Mediterranean (MD) diet is traditionally high in fruits, vegetables, legumes, cereals and extra virgin olive oil. The efficiency of the proteostasis systems, autophagy, notably macroautophagy, and ubiquitin proteasome system (UPS), decline with ageing, with consequent accumulation of harmful oxidized proteins and damaged mitochondria [32]. Although autophagy and UPS have been considered for a long-time independent mechanisms, a growing body of evidence indicates an intimate crosstalk and cooperation between both pathways [33]. Under AD conditions, a close relation does exist between mitochondria dysfunction and cell proteostasis systems (UPS and autophagy), also responsible of the digestion of damaged organelles. Therefore, it is not surprising that mitochondria are considered key players involved in normal aging and the onset of neurodegenerative diseases, notably AD [34]. The proposed mitochondrial cascade hypothesis indicates that mitochondrial changes result in increased Aβ production with ensuing Aβ deposit build-up [35]. Therefore, all these aspects are promising targets to prevent ageing and, possibly, to treat several pathologies, including AD. Our cellular model of AD confirmed the alteration of mitochondria homeostasis after cell exposure to toxic Aβ oligomers, as indicated in Figure 5, which warrants its appropriateness for our study.

Autophagy dysfunctions may contribute to many different pathophysiological conditions. Substantial evidence indicates that senile plaques, a key histopathological hallmark of AD resulting from accumulation of Aβ_1–42_ aggregates, are closely related to the alteration of the autophagic pathway [36]. Moreover, several studies indicate that Aβ_1–42_ influences the expression and activation of a number of proteins involved in autophagy regulation, including p62, mTOR and Beclin1, with an accumulation of autophagosomes in the presence of reduced degradative activity [23]. These data are confirmed in our AD cell model. Indeed, in cells exposed to Aβ_1−__42_ oligomers we observed an increase of Beclin1 expression, of the LC3II/I ratio and, more importantly, of p62 expression, indicating autophagosomes accumulation with impaired degradative activity (Figure 3). p62 accumulation was also confirmed by confocal immunofluorescence shown in Figure 6, with an increase of red signals in cells treated with oligomers.

p62 is remarkably involved in AD onset [37] not only for being a marker of autophagy dysregulation but also of UPS alteration. p62 plays a central role in the crosstalk between UPS and autophagy: it promotes the autophagic degradation by directly binding to the autophagy marker LC3, but it is also involved in the proteasomal degradation of ubiquitinated proteins by its ability to bind to these cargoes [30]. It can also shuttle between the nucleus and the cytoplasm where it binds ubiquitinated cargoes facilitating nuclear and cytosolic protein quality control [30]. Finally, p62 is upregulated and phosphorylated following UPS inhibition, which can facilitate the degradation of ubiquitinated cargoes via autophagy. Differently from the UPS, autophagy can degrade a much wider spectrum of mostly bulkier substrates, such as protein complexes, oligomers and aggregates, and even whole cellular organelles, such as mitochondria.

In our cell model we observed that the MIX prevented the impairment of autophagy and mitochondria functionality induced by Aβ_1–42_ oligomers. Indeed, we obtained a significant activation of the autophagy path in SH-SY5Y cells, as shown by the western blotting analysis of the autophagy markers (Figure 2). In this case, autophagy activation was more stable (the signal was present also after 24 h from treatment) than that previously reported in cells treated only with OleA. In fact, in that case, the highest activation of autophagy was observed after 4 h from treatment, followed by a progressive decline [18]. The efficiency of autophagic degradation under these conditions was confirmed by the significant time-dependent reduction of p62 levels and Aβ_1–42_ oligomers on the surface (Figure 4A) of cells pre-treated with the MIX before exposing them to the oligomers.

The mTOR kinase is a main suppressor of autophagy (following activation through AKT and PI3K signaling) and its negative regulation through AMPK phosphorylation results in activation of autophagy [38]. AMPK also regulates ULK1 kinase, an inhibitor of mTOR, by direct phosphorylation at S555, with ensuing activation of autophagy [38]. In this context, we found that the phosphorylation level of the ribosomal protein S6, a key downstream substrate of TOR, was reduced in cells treated with the MIX for up to 24 h, as compared to untreated control cells (Figure 1B). We also highlighted that, in the same cells, phosphorylation ULK1 at S555 was increased, particularly after 4 h of treatment with the MIX (Figure 1C). These data confirm the involvement of the AMPK pathway in autophagy activation in our cell model and agree with previous results concerning autophagy activation in cells treated with by the sole OleA [18].

Our data indicated that oligomers and fibrils displayed different cytotoxicity; in fact, we did not obtain the same positive results with fibrillar aggregates that, contrary to the oligomers, were present on the surface of cells pre-treated with the MIX. Anyway, we should also consider that previous data indicated that extracellular fibrillar aggregates are digested by the phagocytic activity of microglia cells and not by the neuronal autophagy [39,40]. Also, we cannot exclude that autophagy induction by the MIX is insufficient to interfere with the insults provided by Aβ fibrils, considering that amyloid fibrils in contact with the cell membrane may leak toxic oligomers [41], which would worsen AD symptoms.

Our data demonstrated that the MIX prevents mitochondria damage and could relieves the injury to the crosstalk between ubiquitin proteasome system and autophagy provided by Aβ oligomers (Figure 6).

In conclusion, our in vitro data highlight a significant preventive effect of the MIX against the insults by oligomeric aggregates to neuronal cells. However, to provide more solid evidence of a therapeutic effects of the MIX to damaged neurons, in terms of recovery of the proteostasis equilibrium, a similar investigation should be carried out in a more complex system such as an animal model. Our knowledge is far from being complete and there is still much to learn about autophagy, its role in AD and the beneficial effects of plant polyphenols for AD prevention and therapy in humans. Nonetheless, it appears evident that therapeutic strategies aimed at enhancing autophagy have the potential to be beneficial in AD and small drug-like molecules such as EVOO polyphenols can contribute to the development and implementation of effective therapies anti-age-related pathologies.

## 4. Materials and Methods

### 4.1. Aβ_1–42_ Aggregation

Aβ_1–42_ solutions were prepared by dissolving the lyophilized peptide (Bachem, Bubendorf, Switzerland) in 100% hexafluoroisopropanol (HFIP) to a 1.0 mM final concentration. After HFIP evaporation over-night at room temperature, the samples were stored at −20 °C until use. Amyloid aggregates were grown by dissolving the peptide (25 μM final concentration) in a 20 mM sodium phosphate buffer, pH 7.4, at 25 °C for different lengths of time, without shaking. These aggregation conditions were compatible with the chemical and physical properties of the polyphenols used in the study. Then, the samples were sonicated for 15 min and centrifuged at 18,000× *g* for 15 min at 4 °C; peptide concentration in the clear supernatant was determined from solution absorbance (ε280 = 1490 mol^−1^cm^−1^). Under these conditions, the Aβ_1–42_ sample aggregated for 24 h was mostly populated by oligomers, whereas fibrils were mainly present in the 72 h-aged samples.

### 4.2. Preparation of Oleuropein Aglycone and Hydroxytyrosol Samples

Oleuropein (Extrasynthese, France) was deglycosilated by treatment with almond β-glucosidase (EC 3.2.1.21, Sigma-Aldrich, St. Louis, Germany), as previously described [42]. Briefly, a 10 mM solution of oleuropein in 310 μL of 0.1 M sodium phosphate buffer, pH 7.0, was incubated with 8.9 I.U. of β-glucosidase overnight at room temperature. Then, the reaction mixture was centrifuged at 18,000× *g* for 10 min to precipitate the aglycone (OleA) and the precipitate was resuspended in DMSO in 100 mM stocks. Complete oleuropein deglycosylation to OleA was confirmed by assaying the glucose released in the supernatant with the Glucose (HK) Assay kit (Sigma-Aldrich, Germany). Stocks of OleA were kept frozen and protected from light and were used within the same day once opened.

HT was purchased from Sigma-Aldrich. The powder was dissolved in an aqueous solution at 100 mM final concentration and stored at −20 °C, as previously reported [43]. For experiments, OleA and HT were mixed (MIX) at different OleA:HT molar ratios maintaining the same overall concentration of 75 μM: 2:1 (50 μM:25 μM), 1:1 (37.5 μM:37.5 μM) and 1:2 (25 μM:50 μM).

### 4.3. Cell Culture

Human neuroblastoma (SH-SY5Y) cells were cultured at 37 °C in complete medium (50% HAM, 50% DMEM, 10% fetal bovine serum, 3.0 mM glutamine, 100 units/mL penicillin and 100 μg/mL streptomycin), in a humidified incubator under 5.0% CO_2_. All materials used for cell culture were from Sigma Aldrich. In all experiments the untreated cells (CTRL) correspond to cells treated with the dilution samples buffer used.

### 4.4. MTT Assay

Cell viability was assessed by the MTT assay optimized for the cell line used in the experiments. Briefly, SH-SY5Y cells were seeded into 96-well plates at a density of 6000 cells/well in fresh complete medium and grown for 24 h. The cells were treated with the different concentrations of OleA and HT for 4 h and 24 h, and at the end of the incubation, the culture medium was removed, and the cells were incubated for 1.0 h at 37 °C in 100 μL of serum-free DMEM without phenol red, containing 0.5 mg/mL MTT. Then, 100 μL of cell lysis solution (20% SDS, 50% N,N-dimethylformamide) was added to each well and the samples were incubated at 37 °C for 2 h to allow complete cell lysis. The absorbance of the blue formazan resulting from MTT reduction was read at 570 nm using a spectrophotometric microplate reader. Final absorption values were calculated by averaging each sample in triplicate after blank (100 μL of MTT solution + 100 μL of lysis solution) subtraction.

### 4.5. ROS Determination

Intracellular reactive oxygen species (ROS) were determined using the fluorescent probe 2′,7′–dichlorofluorescein diacetate, acetyl ester (CH2-DCFDA; Thermo-Fisher, Italy), a cell-permeant indicator that becomes-fluorescent in the presence of ROS following removal of the acetate groups by cellular esterases. Product oxidation can be detected by monitoring the increase in fluorescence at 538 nm. SH-SY5Y cells were plated on 96-well plates at a density of 6000 cells/well and exposed for 24 h to the aggregates. In the presence or in the absence of the MIX. Then, 10 μM DCFDA in DMEM without phenol red was added to each well. The fluorescence values at 538 nm were detected after 30 min by Fluoroscan Ascent FL (Thermo-Fisher, Waltham, Italy).

### 4.6. Western-Blotting

SH-SY5Y cells (105 cells/well) were plated in a 6-well plate for 24 h. Following the different treatments with the MIX at different times, the cells were washed with PBS and then lysed in 100 μL of 1× Laemmly buffer (62.5 mM Tris-HCl buffer, pH 6.8, 10% (*w*/*v*) SDS, 25% (*w*/*v*) glycerol) without bromophenol blue. Whole cell lysates were collected and boiled at 95 °C for 5 min and then centrifuged at 12,000× *g* for 5 min at 4 °C. Total protein concentration in lysates was measured by the BCA protein assay kit. β-mercaptoethanol and bromophenol blue were added to an equal amount of protein (20 μg) from each sample, whose components were separated in precast SDS-PAGE gels (Biorad #456-8096) and then transferred onto nitrocellulose membrane by Trans-Blot Turbo Transfer Pack (Biorad #1704157). The immunoblots were incubated at room temperature in PBS containing 5.0% (*w*/*v*) bovine serum albumin, 0.1% (*v*/*v*) tween 20 and probed with primary and appropriate secondary antibodies. The antibodies used in immunoblotting were specific for p-ULK S555 (Merck-Millipore, Burlington, Germany), ULK1 (GeneTex, Irvine, CA, USA), Beclin-1 (Euroclone, Italy), LC3I-II (Euroclone, Italy), p62 (Abcam, UK), phospho-S6 (Euroclone, Italy), S6 (Euroclone, Italy), α-tubulin (Euroclone, Italy), β-actin (Santa-Cruz, Dallas, TX, USA). Finally, the membranes were repeatedly washed in 0.5% (*v*/*v*) PBS- Tween^®^-20 solution and protein bands were detected using the Clarity Western ECL solution. Chemiluminescent signals were acquired by using AmershamTM 600 Imager imaging system (GE Healthcare Life Science, Pittsburgh, PA, USA; the densitometric analysis was carried out using the Quantity One software (4.6.6 version, Bio-Rad, Hercules, CA, USA).

### 4.7. Autophagosome Detection

The induction of autophagy by MIX was monitored by using the Cyto-ID^®^ Autophagy Detection Kit (Enzo Life Sciences, Shanghai, China) in accordance with the manufacturer’s instructions. The Cyto-ID^®^ dye selectively labels the autophagic vacuoles in living cells. The SH-SY5Y cells were seeded on sterilized glass coverslips in a 24-well plate for 24 h; after exposure to MIX for different lengths of time (0.5 h, 1 h, 4 h, 6 h, 24 h), the cells were washed twice with PBS and then with 100 μL of 1× assay buffer provided by the detection kit. Then, the cells were incubated for 30 min at 37 °C with 100 μL of detection reagent (prepared by diluting 1000× the Cyto-ID^®^ Green Detection Reagent in a mixture of 1× assay buffer). After cell fixation with 2.0% (*v*/*v*) paraformaldehyde and three washes with 1× assay buffer, the coverslips were placed on microscope slides using a Fluoromount™ Aqueous Mounting Medium (Merck-Millipore, Burlington, Germany). Sample fluorescence was detected at 488 nm emission using a Leica TCS SP8 scanning microscope (Leica, Mannheim, Germany) equipped with a HeNe/Ar laser source. The observations were performed using a Leica HC PL Apo CS2 X63 oil immersion objective. Cells from three independent experiments and three different fields (about 20 cells/field) per experiment were analysed.

### 4.8. Mitochondrial Membrane Potential

SH-SY5Y cells (3 × 104 cells/well), grown on glass coverslips, after appropriate treatments were incubated at 37 °C for 45 min with 500 nM MitoTracker CMXRos (Thermo Fisher, Italy), Hoechst-33342 nuclear stain and CTX-B Alexa488 for ganglioside, GM1, staining. Then, the cells were fixed in 2.0% buffered paraformaldehyde for 10 min and washed twice in PBS. Coverslips were placed on microscope slides using a Fluoromount™ Aqueous Mounting Medium (Sigma Aldrich-Merck) and multicolor images were collected using a Leica TCS SP8 scanning microscope (Leica, Mannheim, Germany) equipped with 63×, 1.4–0.6 NA, oil, HCX Plan APO lens. The images were acquired using the Leica LAS-AF image acquisition software. Photo montages were generated using the FiJi software, version 8.

### 4.9. Quantification of Active Mitochondria

SH-SY5Y cells, grown for 24 h on 96-well plates (6 × 103 cells/well) in fresh complete medium, were treated with the MIX for 24 h and then for further 24 h with 2.5 μM Aβ_1-42_ oligomers. The mitochondrial membrane potential was assessed by tetraethylbenzimidazolylcarbocyanine iodide (JC-1), a cationic dye that accumulates in energized mitochondria. JC-1 is predominantly a monomer with green fluorescence emission (530 nm). At high mitochondrial membrane potential, the dye aggregates with a red emission (590 nm). According to the provider instructions, the differently treated cells were washed with PBS and incubated for 10 min at 37 °C. Before detection, the cells were washed in PBS and analysed by a fluorescent microplate reader (Biotek Synergy 1H plate reader). The results are reported as red to green fluorescence ratios as compared to untreated cells (CTRL).

### 4.10. Immunofluorescence

SH-SY5Y cells (3 × 104 cells/well) grown on glass coverslips and subjected to the different treatments were washed with PBS. GM1 labelling was performed by incubating the cells with 10 ng/mL CTX-B Alexa488 in complete medium for 10 min at room temperature. Then, the cells were fixed in 2.0% buffered paraformaldehyde for 10 min and permeabilized by treatment with a 1:1 acetone/ethanol solution for 4.0 min at room temperature, washed with PBS and blocked with PBS containing 0.5% BSA and 0.2% gelatin. After incubation for 1.0 h at room temperature with rabbit anti-Aβ_1–42_ polyclonal antibody diluted 1:600 in blocking solution or with 1:500 diluted mouse anti-p62 polyclonal antibody, the cells were washed with PBS for 30 min under stirring and then incubated with Alexa568-conjugated anti-rabbit or Alexa546-conjugated anti-mouse secondary antibodies (Thermo-Fisher, Italy) diluted 1:200 and 1:100 in PBS, respectively. Finally, the cells were washed twice in PBS and once in distilled water to remove non-specifically bound antibodies. Coverslips were placed on microscope slides using a Fluoromount™ Aqueous Mounting Medium (Sigma Aldrich-Merck). Multicolor images were collected using a Leica TCS SP8 scanning microscope (Leica, Mannheim, Germany) equipped with 63×, 1.4–0.6 NA, oil, HCX Plan APO lens. The images were captured using the Leica LAS-AF image acquisition software. Photo montages were generated using the FiJi software, version 8. FRET analysis was performed by adopting the FRET sensitized emission method, as previously reported [44].

### 4.11. Statistical Analysis

Data are reported as mean ± standard error of the triplicate values of at least three independent experiments. Unless otherwise specified, the statistical analysis of the data was performed using the one-way analysis of variance (ANOVA) and pairwise comparisons were performed using Tukey HSD method. Western-blotting statistical analysis were performed by the Kruskal-Wallis test.

## Figures and Tables

**Figure 1 ijms-22-07225-f001:**
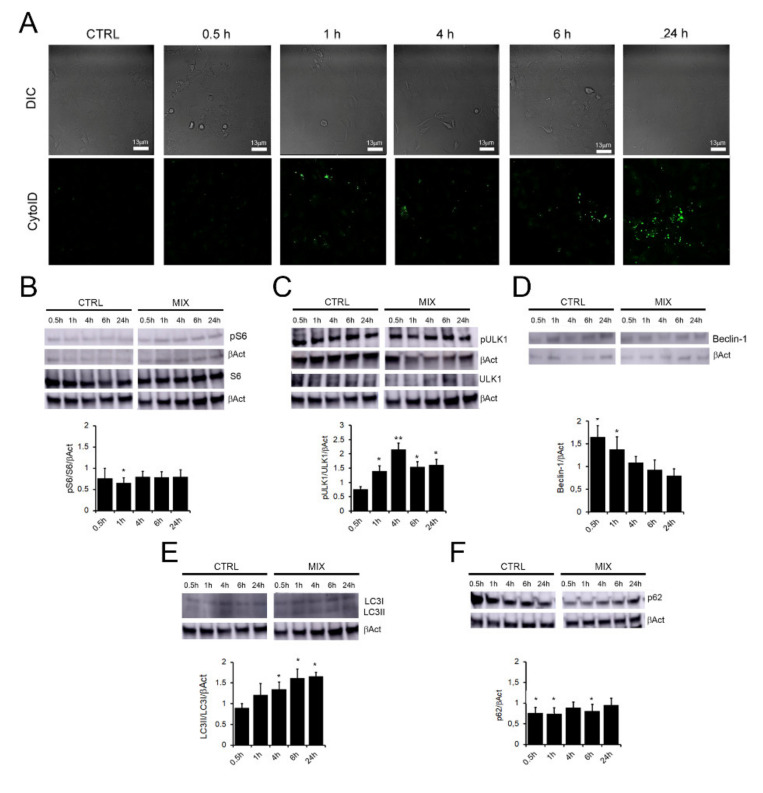
Autophagy detection in SH-SY5Y cells exposed to the MIX. (**A**) SH-SY5Y cells were treated with the OleA:HT mixture at the 1:1 molar ratio (37.5 μM: 37.5 μM) (MIX) for different lengths of time (0.5 h to 24 h). The autophagosomes (green) were labelled with the Cyto-ID^®^ fluorescent dye and the cells imaged by DIC transmission. (**B**–**F**): representative Western blots of all the assayed protein markers of autophagy: (**B**) p-S6/S6tot, (**C**) pULK/ULK, (**D**) Beclin-1 (**E**) LC3II/LC3I ratio and (**F**) p62. Quantification of signals was determined by densitometric analysis of at least three independent experiments normalized on βActin (βAct) signals. All the blot signals were normalized with the control untreated SH-SY5Y cells (CTRL). Error bars represent standard errors. *: *p*-value < 0.05; ** *p* < 0.01 vs. control untreated cells.

**Figure 2 ijms-22-07225-f002:**
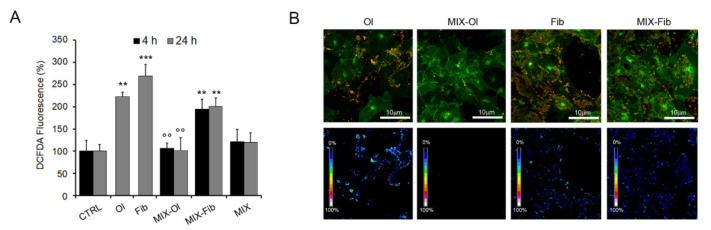
Polyphenols MIX effects on Aβ_1–42_ cells sufferance. (**A**) ROS levels in SH-SY5Y cells treated for 24 h with 2.5 μM Aβ_1–42_ oligomers (Ol) or fibrils (Fib) in absence or after pre-treatment with the 75 μM MIX (MIX-Ol, MIX-Fib) for 4 h (black bars), 24 h (grey bars) assessed by CH2-DCFDA probe. DCFDA fluorescence is reported as percentage respect to untreated control cells. Error bars indicate the standard error of three independent experiments carried out in triplicate. **: *p*-value < 0.01; ***: *p*-value < 0.001 vs. untreated control cells. °°: *p* < 0.01 vs. Aβ_1–42_ aggregates-treated cells. (**B**) Confocal microscopy showing the co-immunolocalization of Aβ_1–42_ aggregates (Ol, Fib) and the membrane GM1 ganglioside in SH-SY5Y cells in absence (Ol, Fib) and in presence of pre-treatment with the MIX for 24 h (MIX-Ol, MIX-Fib). The cells were stained with Alexa 488-conjugated CTX-B probe (green staining); Aβ_1–42_ aggregates were stained with anti-Aβ_1–42_ primary antibody and then with Alexa 568-conjugated anti-rabbit secondary antibody (red fluorescence). FRET efficiency is shown in bottom panels.

**Figure 3 ijms-22-07225-f003:**
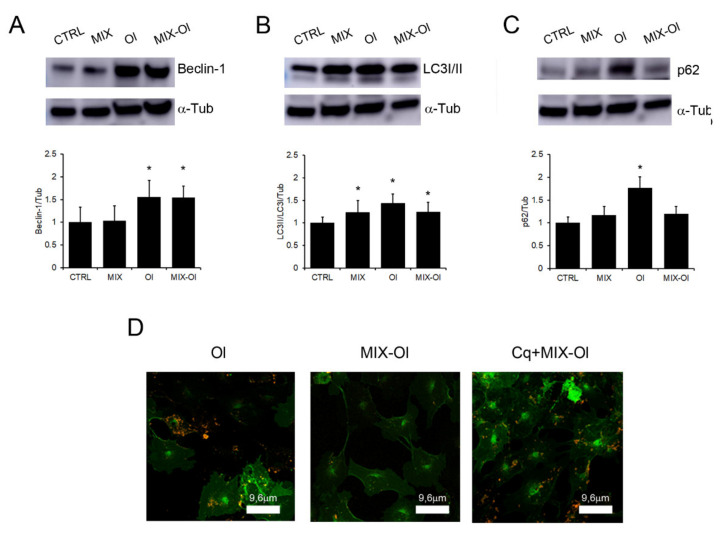
MIX-mediated recovery of redox homeostasis in human neuroblastoma cells depends on autophagosome–lysosome fusion. (**A**–**C**) The SH-SY5Y cells were treated with (i.) 75 μM OleA/HT (MIX) for 4 h; (ii.) Aβ_1–42_ oligomers (Ol); (iii.) MIX for 24 h followed by exposure to oligomers for further 24 h (MIX-Ol). Western blots of all the assayed protein markers of autophagy: (**A**) Beclin 1, (**B**) LC3I/LC3II ratio; (**C**) p62. Quantification of signals was determined by densitometric analysis of at least three independent experiments normalized on α−Tubulin (α−Tub) signals. All the blot signals were normalized with the control untreated SH-SY5Y cells (CTRL). Error bars represent standard errors. *: *p*-value < 0.05 vs. control untreated cells. (**D**) SH-SY5Y cells were treated with (i.) 2.5 μM oligomers (Ol) for 24 h; (ii.) MIX (75 μM) for 24 h and then with oligomers for further 24 h (MIX-Ol); (iii.) CQ (10 μM) for 16 h followed by different treatments (CQ+MIX-Ol). Immunolocalization of Ol on the plasma membrane by confocal microscopy. The cells were stained with Alexa 488-conjugated CTX-B (green fluorescence); Aβ_1–42_ Ol aggregates were stained with anti- Aβ_1–42_ antibodies and with Alexa 568-conjugated anti-rabbit secondary antibodies (red fluorescence).

**Figure 4 ijms-22-07225-f004:**
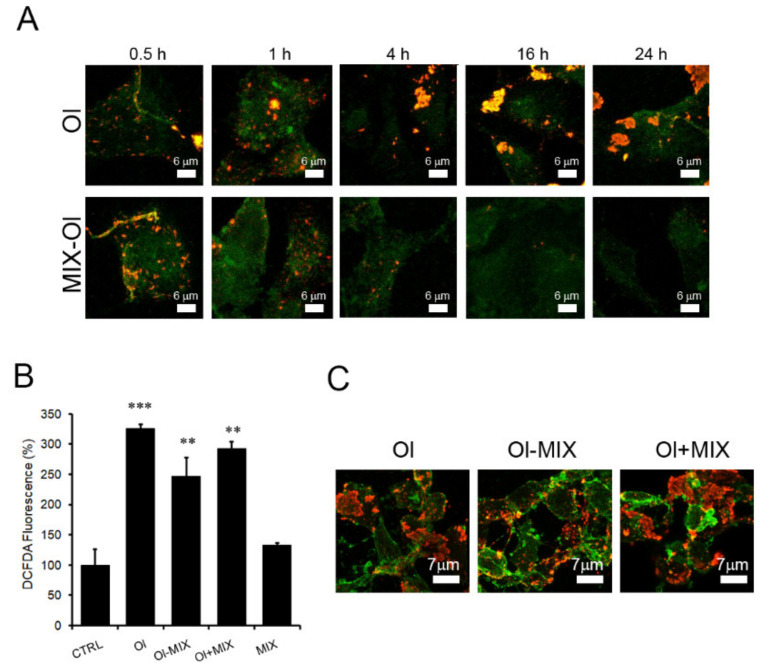
The MIX does not alleviate aggregate-impaired autophagy. Aggregate-GM1 co-localization and ROS levels in SH-SY5Y cells treated with oligomers (2.5 μM, monomer concentration) for different times (0.5 h to 24 h) in absence or in presence of a 24 h pre-treatment with MIX. (**A,C**) Immunolocalization of Aβ_1–42_ oligomers on the plasma membrane by confocal microscopy. Cells were stained with Alexa 488-conjugated CTX-B (green fluorescence); Aβ_1–42_ oligomers were stained with anti- Aβ_1–42_ antibodies and with Alexa 568-conjugated anti-rabbit secondary antibodies (red fluorescence). (**B**) intracellular ROS levels were detected by CH2-DCFDA probe in SH-SY5Y cells exposed to oligomers (2.5 μM) for 24 h followed by treatment with the MIX for further 24 h (Ol-MIX), or co-treated for 24 h with the MIX and Aβ_1–42_ oligomers (Ol+MIX DCFDA fluorescence are reported as percentage respect to untreated control cells. Error bars refer to the standard errors of at least three independent experiments. **: *p*-value < 0.01; ***: *p*-value < 0.001 vs. untreated control cells.

**Figure 5 ijms-22-07225-f005:**
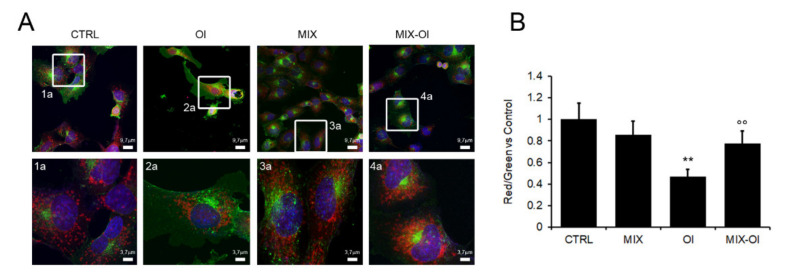
The MIX protects mitochondria functionality. SH-SY5Y cells were pre-treated with the 75 μM MIX for 24 h and then exposed for further 24 h to 2.5 μM (monomer concentration) Aβ_1–42_ oligomers (Ol). (**A**) The cells were stained with MitoTracker CMXRos, Hoechst-33342 Nuclear stain and CTX-B Alexa488 for GM1 staining. Magnification is shown in panels. (**B**) Mitochondria functionality, as assessed by quantification of JC1 signals expressed as Red/Green ratio. Error bars refer to the standard errors of at least three independent experiments. **: *p*-value < 0.01 vs. untreated control cells; °°: *p*-value < 0.01; vs. Aβ_1–42_ aggregates-treated cells.

**Figure 6 ijms-22-07225-f006:**
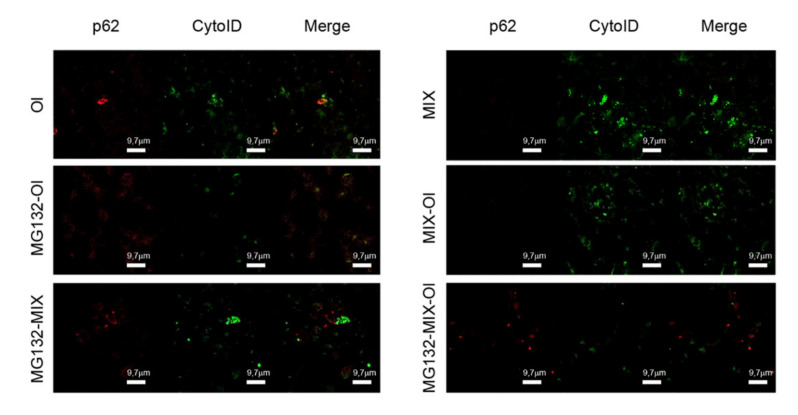
The MIX effects on p62 activity. Autophagosome formation in SH-SY5Y cells treated with (i.) oligomers (Ol) (2.5 μM) for 24 h; (ii.) MIX (75 μM) for 24 h and then with oligomers for further 24 h (MIX-Ol); (iii.) MG132 (5.0 μM) for 6 h before different treatments. Autophagosomes were stained by Cyto-ID^®^ fluorescent dye (green fluorescence); p62 was stained with anti-p62 antibodies and Alexa 564-conjugated anti-mouse secondary antibodies (red fluorescence).

## Data Availability

Not Applicable.

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
