# Peer review of "EVOO Polyphenols Relieve Synergistically Autophagy Dysregulation in a Cellular Model of Alzheimer’s Disease"

_ijms, 2021, doi:10.3390/ijms22137225_

Round 1
Reviewer 1 Report
The authors describe the effects of a MIX on neuronal cells viability and autophagy and the ability of the MIX to counteract the detrimental effects of amyloid oligomers. The reviewer thinks that the paper needs major revisions in order to be accepted.
1) The authors describe the protocol for amlyloid aggregates formation. However, they do not show any data on the effective amyloid aggregation state used to treat cells. They should give us a western blotting for example, evidencing oligomers and fibrillary structures.
2) Oleuropein and its derivative are dissolved in different solvents. How the author maintained the same amount of DMSO in the different MIX (1:1, 2:1 and 1:2)?
3) Performing the experiments with the 1:1 MIX, what is included in the control?
4) Western blotting are of very poor quality and in some figures it is very hard to give them an interpretation. Why all the images start with 0,5h and not with time 0? When 0,5 h is marked with an asterisk (Fig.1), with respect to which time point is it significant? Time 0h and untreated cells point is never shown.
5) In order to get an idea of the crosstalk autophagy-proteasome, it could be interesting to monitor proteasome activity in cell lysates treated with both the mix and oligomers, otherwise is not possible to assess "suggesting that protection by MIX also extended to the UPS".
6) English language should be extensively revised, there are sentences not easy to follow.
Author Response
Dear Ms. Tinsley Qiu
Assistant Editor,
Thank you for evaluating our Manuscript ID: ijms-1223029
Titled: EVOO polyphenols relieve synergistically autophagy dysregulation in a
cellular model of Alzheimer's disease
Authors: Manuela Leri, Andrea Bertolini, Massimo Stefani *, Monica Bucciantini
We thank the referees for their appreciation and thoughtful critiques, which we took into account to modify the manuscript. We feel the quality of our study has been improved and we hope that now our manuscript can be considered suitable for publication in your journal.
Best regards,
Prof. Massimo Stefani
Review1:
Comments and Suggestions for Authors
The authors describe the effects of a MIX on neuronal cells viability and autophagy and the ability of the MIX to counteract the detrimental effects of amyloid oligomers. The reviewer thinks that the paper needs major revisions in order to be accepted.
- The authors describe the protocol for amlyloid aggregates formation. However, they do not show any data on the effective amyloid aggregation state used to treat cells. They should give us a western blotting for example, evidencing oligomers and fibrillary structures.
Accordingly, with previous published results (doi: 10.1016/j.fct.2019.04.015) we analyzed our samples by ThT fluorescence (Fig.1A) and electron microscopy (Fig.1A) in order to asses the amyloid features of our aggregates. The figure is reported in word file uploaded.
- Oleuropein and its derivative are dissolved in different solvents. How the author maintained the same amount of DMSO in the different MIX (1:1, 2:1 and 1:2)?
We used different dilution of OleA in order to obtain the same amount of DMSO in all ratio, and all OleA samples are diluted in the same buffer of HT. We prepared OleA 100x for each dilution used. Moreover, each ratio has as controls cells treated with the dilution samples buffer.
- Performing the experiments with the 1:1 MIX, what is included in the control?
The control is untreated cells corresponding to cells treated with the dilution sample buffer. We added the sentence in material and methods section.
- Western blotting are of very poor quality and in some figures it is very hard to give them an interpretation. Why all the images start with 0,5h and not with time 0? When 0,5 h is marked with an asterisk (Fig.1), with respect to which time point is it significant? Time 0h and untreated cells point is never shown.
We improved the quality of western-blotting images. The blots are compared with untreated cells reported for each time in supplementary figure 3.
- In order to get an idea of the crosstalk autophagy-proteasome, it could be interesting to monitor proteasome activity in cell lysates treated with both the mix and oligomers, otherwise is not possible to assess "suggesting that protection by MIX also extended to the UPS".
We agree with the reviewer but unfortunately now we are not able to improve this aspect, so we modify the manuscript.
- English language should be extensively revised, there are sentences not easy to follow.
We extensively revised the text.

Reviewer 2 Report
The study by Leri et al describes the very thorough and extensive investigation of the benefit of EVOO consitituents for autophagy flux prior to Abeta administration. I believe this paper not only provides some evidence for benefits of EVOO but also provides valuable insights into the mechanisms of action of Abeta oligomers. I do have a number of suggestions and comments for further improvement of the manuscript.
1) Please check the figure legend labelling against references in text. I noticed that figure 3A and B were swapped. There maybe other issues I did not pick up.
2)page 3, line 104: please revise "not-enough preventative effect"
3)Stats are generally a bit unclear. To me at least, I am not certain exactly what conditions are being compared when a star above a condition is shown. I can understand when there is a control but for figure 1 when there are different times? It is not clear to me.
Some are confusing - for example, Fig 1C, 6h and 24h look almost identical but both have a single star above.
In the text, Beclin is referred to as 1B, but this is labelled pS6? Beclin is shown in the graph in 1D but it appears to decrease not increase. Major errors here! Overall, the data shown here is quite weak and needs attention. Only pULK appear to support an increase in autophagy over time to 6 h?
4) The studies include ROS production and binding to GM1. These are interesting. However, it has been previously shown that MTT assays are hampered by artifacts in the presence of amyloid fibrils in particular (Abeta) because of an increase in the crystals due to the fibrils. It is essential therefore that the MTT assays are replaced with an alternative toxicity assay or removed entirely. Ideally, a Live Dead, or cell count would be used which avoids the issues with artefacts due to the aggregates. It is very possible that the increased toxicity measured here for fibrils is due to this issue.
E.g https://pubmed.ncbi.nlm.nih.gov/12689605/
5) The methods for Abeta preparation include a step using HFIP. HOw do the authors ensure removal of the HFIP and check it is entirely removed?
6) in the methods, the authors mention that 24h produces oligomers and while 72h fibrils but can they cite evidence for this? Did they check?
Overall, I think there are some really intreresting and important findings here but there are some issues regarding the organisation of the data and figures. And the cytotoxicity assays should be replaced or removed.
Author Response
Dear Ms. Tinsley Qiu
Assistant Editor,
Thank you for evaluating our Manuscript ID: ijms-1223029
Titled: EVOO polyphenols relieve synergistically autophagy dysregulation in a
cellular model of Alzheimer's disease
Authors: Manuela Leri, Andrea Bertolini, Massimo Stefani *, Monica Bucciantini
We thank the referees for their appreciation and thoughtful critiques, which we took into account to modify the manuscript. We feel the quality of our study has been improved and we hope that now our manuscript can be considered suitable for publication in your journal.
Best regards,
Prof. Massimo Stefani
Review 2:
The study by Leri et al describes the very thorough and extensive investigation of the benefit of EVOO consitituents for autophagy flux prior to Abeta administration. I believe this paper not only provides some evidence for benefits of EVOO but also provides valuable insights into the mechanisms of action of Abeta oligomers. I do have a number of suggestions and comments for further improvement of the manuscript.
- Please check the figure legend labelling against references in text. I noticed that figure 3A and B were swapped. There maybe other issues I did not pick up.
We correct the mistake.
- page 3, line 104: please revise "not-enough preventative effect"
We correct the typing error.
- Stats are generally a bit unclear. To me at least, I am not certain exactly what conditions are being compared when a star above a condition is shown. I can understand when there is a control but for figure 1 when there are different times? It is not clear to me. Some are confusing - for example, Fig 1C, 6h and 24h look almost identical but both have a single star above. In the text, Beclin is referred to as 1B, but this is labelled pS6? Beclin is shown in the graph in 1D but it appears to decrease not increase. Major errors here! Overall, the data shown here is quite weak and needs attention. Only pULK appear to support an increase in autophagy over time to 6 h?
Cell lysates collected from treated cells were compared to the respective untreated cells for each time point. All controls are shown in supplementary file, Figure 3.
4) The studies include ROS production and binding to GM1. These are interesting. However, it has been previously shown that MTT assays are hampered by artifacts in the presence of amyloid fibrils in particular (Abeta) because of an increase in the crystals due to the fibrils. It is essential therefore that the MTT assays are replaced with an alternative toxicity assay or removed entirely. Ideally, a Live Dead, or cell count would be used which avoids the issues with artefacts due to the aggregates. It is very possible that the increased toxicity measured here for fibrils is due to this issue.
E.g https://pubmed.ncbi.nlm.nih.gov/12689605/
We agree with the reviewer for his/her comment, and for this reason we have always paid great attention to evaluate any interference in our systems. In order to observe interference induced by fibrillar or oligomeric species on MTT reduction, we performed MTT assay on cells pre-treated with aggregates: oligomers (Ol) or fibrils (Fib) and parallelly in other cells MTT was administered together with pre-formed aggregated (Ol+MTT, Fib+MTT). The results, shown in figure 1, indicate that when amyloid species are added together MTT in cellular medium we did not observe any decrease of MTT reduction suggesting the absence of any interference in our model. The figure1 is reported in the uploaded.
- The methods for Abeta preparation include a step using HFIP. HOw do the authors ensure removal of the HFIP and check it is entirely removed?
The HFIP used to dissolve the Ab stock solutions was evaporated overnight before the use of the peptide.
- in the methods, the authors mention that 24h produces oligomers and while 72h fibrils but can they cite evidence for this? Did they check?
Accordingly with previously published data (doi: 10.1016/j.fct.2019.04.015) we check the amyloid aggregation path of Ab through ThT fluorescence (Fig. 2B) and electron microscopy (Fig. 2A). The figure2 is reported in the uploaded file.
Overall, I think there are some really intreresting and important findings here but there are some issues regarding the organisation of the data and figures. And the cytotoxicity assays should be replaced or removed.

Round 2
Reviewer 1 Report
The paper has been improved by the authors but some problems are still evident.
The authors should add a CTRL point to the western images of Fig. 1, as performed for the image in panel A. In addition, the WB of S1, showing control untreated cells, should be moved to fig. 1, otherwise is not possible to understand the comparison between treated and control cells.
In fig 3, there is no reason to add coloured MWM to the images without any indications of the Molecular weight. Panel C has a WB in a different format with respect to panels A and B. They should be shown in the same format.
Author Response
The authors should add a CTRL point to the western images of Fig. 1, as performed for the image in panel A. In addition, the WB of S1, showing control untreated cells, should be moved to fig. 1, otherwise is not possible to understand the comparison between treated and control cells.
We changed the Fig.1: we moved in Fig.1 the controls untreated cells.
In fig 3, there is no reason to add coloured MWM to the images without any indications of the Molecular weight. Panel C has a WB in a different format with respect to panels A and B. They should be shown in the same format.
We changed Fig.3: we removed the colored MWM
We used the same format for panel A, B and C
Reviewer 2 Report
I did not feel that the authors considered my comments fully and the issues remain unresolved regarding toxicity assays using MTT. I would urge the authors to add additional toxicity assays to complete their study
Author Response
I did not feel that the authors considered my comments fully and the issues remain unresolved regarding toxicity assays using MTT. I would urge the authors to add additional toxicity assays to complete their study
We apologize for giving the impression that we have not taken into consideration the reviewer comments. We thank and respect her/his decisions. During the refereeing, we felt compelled to show the controls we normally do to make sure of any interference and to avoid misinterpretation of the data. Furthermore, since, in our model, the cells treatments do not induce cell death but only cellular suffering, it is probable that the use of a test that discriminates live cells from dead cells could give different results from those obtained with an MTT test, which instead highlights a state of cellular suffering even in the absence of cell death. Anyway, according with the reviewer suggestion, we removed the MTT assays from the paper.
Round 3
Reviewer 1 Report
The reviewer thinks that the authors correctly followed the suggestions.